# Recent Advances in Hormonal Regulation and Cross-Talk during Non-Climacteric Fruit Development and Ripening

**Lida Fuentes** [1,*]**, Carlos R. Figueroa** [2] **and Monika Valdenegro** [3]

[1] Centro Regional de Estudios en Alimentos Saludables (CREAS), CONICYT-Regional GORE Valparaíso Proyecto R17A10001, Avenida Universidad 330, Placilla, Curauma, Valparaíso 2340000, Chile

[2] Institute of Biological Sciences, Campus Talca, Universidad de Talca, Talca 3465548, Chile; cfigueroa@utalca.cl

[3] Escuela de Agronomía, Facultad de Ciencias Agronómicas y de los Alimentos, Pontificia Universidad Católica de Valparaíso, Calle San Francisco s/n, Quillota 2260000, Casilla 4-D, Chile; monika.valdenegro@pucv.cl

\* Correspondence: lfuentes@creas.cl; Tel.: +56-32-237-2868

**Abstract:** Fleshy fruits are characterized by having a developmentally and genetically controlled, highly intricate ripening process, leading to dramatic modifications in fruit size, texture, color, flavor, and aroma. Climacteric fruits such as tomato, pear, banana, and melon show a ripening-associated increase in respiration and ethylene production and these processes are well-documented. In ontrast, the hormonal mechanism of fruit development and ripening in non-climacteric fruit, such as strawberry, grape, raspberry, and citrus, is not well characterized. However, recent studies have shown that non-climacteric fruit development and ripening, involves the coordinated action of different hormones, such as abscisic acid (ABA), auxin, gibberellins, ethylene, and others. In this review, we discuss and evaluate the recent research findings concerning the hormonal regulation of non-climacteric fruit development and ripening and their cross-talk by taking grape, strawberry, and raspberry as reference fruit species.

**Keywords:** fruit quality parameters; phytohormones; ethylene; auxin; abscisic acid (ABA); brassinosteroids; jasmonic acid; grape; strawberry; raspberry

## 1. Introduction

Angiosperms produce different fruit categories that evolved to best suit seed protection and aid in seed dispersal towards the final stages of fruit ripening. Botanically, fleshy fruits are very varied in the way they develop. For example, in grape (*Vitis* spp.) and tomato (*Solanum lycopersicum* L.), fruit is from the result of a developed ovary, whereas, in strawberry (*Fragaria* spp.), apple (*Malus* × *domestica* Borkh.), and pineapple [*Ananas comosus* (L.) Merr.], it results from the accessory tissue external to the carpels, the receptacle [1,2], or formed from a series of ovaries (drupelets) attached to a receptacle as is the case of fruits belonging to the genus *Rubus* [3–10]. Evolutionary studies have revealed that both dry and fleshy fruit share some common developmental mechanisms, and that fleshy fruit has evolved from ancestral dry fruit-producing species [11]. Whether fleshy or dry fruit, the final fruit involves a progression of specific steps, namely, fruit set, fruit development, and fruit ripening and senescence [2,12]. In general, fruit ripening is marked by very important phase changes that result in the conversion of less appetizing green fruit into a highly palatable, aromatic, colored, and nutritionally rich fruit.

Fruits are characterized by having a developmentally and genetically controlled development and ripening process, leading to both physiological and biochemical changes [1,13–17]. These changes generally occur through tightly regulated events such as, (i) colour change, by the decrease of chlorophyll content, with anthocyanin and/or carotenoid accumulation, (ii) changes in modification of cell wall structure and cell turgor that result in the change of fruit texture, (iii) increases of different metabolites (acids, sugars, volatiles, among others) that impact flavor, aroma, and nutritional quality, and (iv) increased susceptibility to pathogen attack at the later stages of ripening that result in fruit spoilage [1,2,14,18].

Based on their ethylene evolution and respiration pattern, fleshy fruits are categorized as climacteric and non-climacteric, according to the regulatory mechanisms underlying the ripening process [19,20]. Climacteric fruits or ethylene-dependent fruits have the capability to ripen after harvest with the help of ethylene production [21]. Climacteric fruits such as banana (*Musa* spp.), tomato, avocado (*Persea americana* Mill.), and apple are characterized by a dramatic increase in respiration and ethylene evolution during the onset of ripening [1,2,13]. Non-climacteric fruits are not capable to ripen after removal from the parent plant, whereas climacteric fruits are [21]. Those fruits like strawberry, grape, raspberry (*Rubus idaeus* L.), and citrus (*Citrus* spp.) are defined by the absence of an ethylene-related respiratory peak and do not show a climacteric rise in ethylene evolution [1,2,13,22]. Despite the above, the classification of fruits as either climacteric or non-climacteric is not obvious, because some fruits, like melons (*Cucumis* spp.), can display both climacteric and non-climacteric behaviors [23]. Furthermore, there are climacteric and suppressed-climacteric plum varieties, whose ability to respond to ethylene could be affected [24]. More complicated regulation has been observed in kiwifruit, where the first stage of ripening is not dependent on ethylene whereas the second stage is [25]. For other fruit such as strawberry and raspberry, typically classified as non-climacteric, current molecular studies suggest a certain role of ethylene in ripening. It was reported that transcript levels of ethylene receptors increase at the onset of ripening in strawberry [26], and transcript levels of ethylene biosynthesis increase with ripening progression in raspberry [10].

Independent of classification as climacteric or not, color and textural changes are the main modifications observed during fruit ripening together with changes of organic acids, sugars, and volatile compounds [12,14] that contribute to fruit flavor, particularly by adjusting the equilibrium between organic acids and sugar [27,28]. Modifications in fruit size and color are considered important parameters for the ripening-stage differentiation of many fruits, including the non-climacteric fruits such as grape, strawberry, and raspberry [10,29–31]. The red or purple color seen in these fruits are mainly due to the accumulation of anthocyanins. These compounds are water-soluble pigments, synthesized as products of the phenylpropanoid pathway in the cytosol and localized in vacuoles during ripening and stress responses, and have been widely studied in different fruit species including raspberry and blackberry, due to their health benefits [32–39].

Postharvest fruit softening contributes to the deterioration of fruit quality and makes postharvest management difficult with fleshy fruit. The fungus *Botrytis cinerea* Pers.:Fr. has often been reported as the main pathogen causing rapid fruit decay, especially in grape, strawberry, and raspberry [38–40]. Cell wall changes accompanied by an intense decrease in the content and the degree of methyl-esterification of pectin during ripening have been reported in different raspberry cultivars [41], Chilean strawberry [*Fragaria chiloensis* (L.) Mill.] and cultivated strawberry (*Fragaria × ananassa* Duch.) [30,42].

Changes in flavor and taste are directly reliant on the sugar–acid balance and contents of the fruit, which is significantly important for consumers [27,28]. For instance, too much acid results in a tart and unpalatable fruit; conversely, too little results in bland and insipid fruit. Acid levels, expressed as titratable acidity (TA), changes in starch breakdown, and soluble sugar increase have been used as indicators of taste [43] and a critical index of fruit ripening [28]. Also, flavour can be determined by the presence of molecules such as anthocyanins and tannins, that grant slight and pleasing astringency [28]. Aroma, one of the most valued attributes of raspberry [44] and strawberry [45,46] fruits, is a parameter that depends on a number of factors such as concentration, combination, and the perception threshold

of different volatile compounds [45–47], and the main aroma components include lipid-derived compounds, phenolic derivatives, amino acid-derived compounds, and mono- and sesquiterpenes [47].

Several studies have indicated that the regulation of development and ripening, and therefore the final quality, of non-climacteric fruit involves the combined action of many phytohormones, such as auxin, abscisic acid (ABA), gibberellins (GAs), ethylene, cytokinins (CKs), brassinosteroids (BRs) and jasmonic acid (JA) [1,2,48–53]. However, when compared with climacteric fruit, the hormonal regulatory mechanisms involved in non-climacteric fruit ripening are not well-characterized compared with climacteric ones. In this review, we discuss recent research on hormonal regulation of non-climacteric fruit development and ripening, and further evaluate their crosstalk.

## 2. Hormonal Regulation of Ripening in Non-Climacteric Fruit

Conventionally, non-climacteric fruits have been classified as a separate group that did not show the typical climacteric ripening pattern. However, studies including comparative genomic data analysis carried out in tomato and hot pepper (*Capsicum* spp.) as climacteric and non-climacteric fruit models, respectively, show that the expression of genes encoding for transcription factors such as non-ripening (NOR), tomato AGAMOUS-like 1 (TAGLI) and ripening inhibitor (RIN), and for ethylene signaling pathway-related components are common steps in both fruit categories [54]. Also, identification of MADS-box genes in these two categories of fruit suggests that at least some molecular regulatory processes of fruit ripening are common between climacteric and non-climacteric fruits [55].

Plant hormones are widely known to be regulators of fruit development and ripening [1–3,56–58]. Recent evidence has indicated that the shared action of three hormones, namely, auxin, cytokinin, and gibberellins, contributes to normal fruit growth even in the absence of fertilization, a process known as parthenocarpy; application of these hormones alone starts fruit development in many species [2,59–63], suggesting that communication between these hormones is necessary for fruit set and fruit growth. An overview of phytohormones involved in non-climacteric fruit development and ripening (especially of grape, strawberry, and raspberry) and their possible crosstalk is described below (Figure 1).

### *2.1. Abscisic Acid (ABA)*

ABA is a versatile phytohormone that controls a broad range of plant behaviors such as the adaption to stress conditions, seed dormancy, seedling growth and fruit development [1,64,65]. In non-climacteric fruits, no significant increase in ethylene occurs, but ABA seems to play a major role during ripening [2,12]. Many years ago, ABA was reported to be involved in strawberry fruit ripening [66] and has been described as the major regulator of grape berry ripening onset [67], including coloration, sugar accumulation, and softening [65,68–70]. In grape and strawberry, the ABA level was reported to be low during early fruit development stages, and high at different fruit ripening stages [48,71]. In strawberry, ABA treatment was reported to accelerate fruit softening, increase colour and ethylene production, and induce phenylalanine ammonia-lyase (PAL) enzyme activity, the first committed step in the phenylpropanoid pathway involved in the biosynthesis of polyphenol compounds, such as flavonoids [1,58].

The application of ABA to strawberry fruit resulted in enhanced anthocyanin content during ripening, which was associated with the upregulation of gene expression related with enzymes for the later steps for anthocyanin biosynthesis, such as anthocyanidin synthase (ANS) and glucosyltransferase (GT) [72]. The same study showed an increased L-ascorbic acid content under ABA-treatment [72]. The improvement in strawberry fruit functional quality, phenolic and anthocyanin compounds, L-ascorbic acid, and antioxidant activity, by mild salt and drought stress was associated with an upregulation of several genes involved in ABA biosynthesis [73]. On the other hand, the promotion of strawberry fruit ripening by sucrose was associated with stimulation of ABA biosynthesis [74]. In 'Flame Seedless' grapes, increased anthocyanin content, improvement in color, and acceleration of fruit softening were observed under exogenous treatment with ABA [75]. All these antecedents indicate that ABA regulates

the accumulation of important metabolites to define a ripe fruit and, in turn, these metabolites have a regulatory effect on the ABA biosynthesis.

Genetic and biochemical studies have indicated that 9-*cis*-epoxycarotenoid dioxygenase (NCED) is the main enzyme in the ABA biosynthetic pathway in both climacteric and non-climacteric fruit [76–78]. In peach and grape, the respective genes -*PpNCED1* and *VvNCED1*- were higher expressed at the onset of fruit ripening, preceding the ABA accumulation, and particularly necessary for the increase in ethylene production during peach ripening [77]. In both fruits, treatment with fluridone (an ABA synthesis inhibitor) produced a significant decrease of ABA levels maintaining the fruit firmness during storage. The importance of the NCED enzyme in strawberry fruit development has also been demonstrated [48,79]. A substantial decrease in ABA content and the absence of red colouration in fruit receptacles was observed under nordihydroguaiaretic acid (NDGA) treatment, a recognized inhibitor of NCED activity [79]. Similar results were observed in strawberry agro-infiltrated with *FaNCED1*-RNAi constructs [48]. On the other hand, the increase of *FaNCED1* mRNA levels and ABA accumulation using a fruit-tissue sucrose-incubation test suggests that *FaNCED1* expression can be regulated by accumulated metabolites, like sucrose, during strawberry ripening [74].

The components of the ABA signal transduction pathway have been elucidated in *Arabidopsis thaliana* (L.) Heynh. [80–82], and two essential ABA signaling pathways have been suggested: the 'ABA-PYR/PYL/RCAR-PP2C-SnRK2' [83], and the 'ABA-ABAR-WRKY40-ABI5' pathways [84]. In the first one, ABA binding to the receptor PYR1 (for pyrabactin resistant) stimulates the interaction with the PROTEIN PHOSPHATASE2C (PP2C), resulting in the inactivation of PP2C and concomitant activation of SUCROSE NONFERMENTING-RELATED KINASE2 (SnRK2). The activation of the latter enzyme turns on ABA signaling through the phosphorylation of downstream targets, such as ABA-response element-binding transcription factors (AREB/ABF), activating the expression of ion channels and NADPH oxidases [80–83,85–87]. The second pathway is initiated by the putative ABA receptor ABAR [19,88,89]. In this pathway, the WRKY40 transcription factor acts as a suppressor of ABA signaling, observing an ABAR–WRKY40 interaction when the ABA becomes high. This ABAR–WRKY40 interaction results in upregulation of ABA-responsive gene expression, including that related with ABA-responsive transcription factors such as ABA Insensitive, ABI5 and ABI4 [84,90–92].

The ABA receptors PYR1 and ABAR have recently been characterized in grape [93,94] and strawberry [19,48,95,96]. In 'Kyoho' grape, the expression level of PYR1-like gene (*VlPYL1*) was highest in fruit and also in leaf tissue, with an increase of this transcript during fruit development, and a consequent reduction in ripe berries [93]. Over-expression of *VlPYL1* improved ABA sensitivity in Arabidopsis. Therefore, the upregulation of the expression of this gene not only induced a set of ABA-responsive gene transcripts, including those encoding for transcription factor ABF2 and β-glucosidases BG3 that are important for ABA biosynthesis, but also improved the quality of the grape berry, promoting anthocyanin increase [93]. In strawberry, transcriptional analysis in achenes and receptacle during ripening was the first antecedent for the role of ABA/PYR pathway, observing differential expression of genes encoding for protein phosphatase 2C (PP2C), RAS-related small GTP binding protein, and putative serine/threonine protein kinase (ARSK1) [97]. Later, it was demonstrated that under ABA stimulation the ABAR promotes ripening in strawberry fruit [19,48,95], through ABA binding [96]. The downregulation of *FaABAR* expression inhibits ripening, without rescued by addition of exogenous ABA, suggesting a crucial role for the ABAR receptor in strawberry ripening [46]. Regarding the FaABAR interaction with other potential signaling proteins, a leucine-rich repeat receptor-like kinase LRR-RLK gene, named as *FaRIPK1* (for red-initial protein kinase 1), has been localized in the nucleus and plasma membrane, and its binding to FaABAR regulates strawberry fruit ripening [94]. During strawberry ripening, the regulatory action of the transcription factor FaMYB10 -an important regulator of anthocyanin synthesis- was regulated through the perception of ABA by FaABAR [95].

The role of AREB/ABF—the most described transcription factor of the ABA pathway, has also been studied in grape berry ripening [65,97]. The expression of *VvABF2*, previously described as *GRIP55* for GRAPE RIPENING-INDUCED PROTEIN 55 [97], is induced by ABA and showed accumulation from véraison until the final phase of the berry ripening in 'Cabernet Sauvignon' cultivar [65]. The same study showed that the grape cells over-expressing *VvABF2* have shown improved responses to ABA treatment, when compared with control cells, affecting the cell wall modification and synthesis of phenolic compounds. Tomato fruits over-expressing *VvABF2* decreased its firmness, indicating that VvABF2 is an important transcription factor that regulates the ABA-dependent pathway during ripening of grape berry [65].

### 2.2. Indole-3-Acetic Acid (IAA)

The critical role of auxin in fruit development has been well established, and indole-3-acetic acid (IAA) was described as the main regulator of fruit development in climacteric and non-climacteric fruit [1,2,98–100]. Several years ago, the importance of auxins for fruit set and subsequent growth was established in tomato [98–100].

In grape, the highest content of IAA has been reported in flowers and young berries, and it gradually decreases to low levels at véraison and throughout the ripening period [101–104]. The negative role of IAA in sugar accumulation and anthocyanin content has been associated with ripening delay of the grape berry [103,104]. In strawberry, achenes have been ascribed as the principal auxin source and promoter for the growth of the strawberry receptacle [31,105,106]. The removal of achenes from unripe receptacles caused the inhibition of growth and expansion, inducing the expression of ripening-related genes [30,31,107–110]. A transcriptional analysis of strawberry fruit exogenously treated with auxin showed downregulation of flavonoid pathway (e.g., chalcone synthase)-, aroma biosynthesis (e.g., alcohol acyltransferase)-, and cell wall modification (e.g., pectate lyase, endo-1,4-beta-glucanase and expansins 1 and 2)-related genes [110]. In raspberry and strawberry, the potential role of IAA in fruit development was demonstrated by the overexpression of the auxin synthesis-related gene (*DefH9-iaaM*) in transgenic plants, that showed increased fruit size and number [111]. However, our results showed no significant differences in firmness, total soluble solids or titratable acidity during in vitro IAA assay of raspberry at the onset of fruit ripening [53]. Thus, there is still much to decipher with regard to IAA's role and relation with other hormones in raspberry.

The synthesis and transport of auxins in non-climacteric fruits is an area that needs further investigation. In Arabidopsis [112–116] it was reported that IAA is synthesized by a two-step pathway, in which, the family of TRYPTOPHAN AMINOTRANSFERASE OF ARABIDOPSIS1/TRYPTOPHAN AMINOTRANSFERASE RELATED (TAA1/TAR) proteins [115,116] transform tryptophan to indole-3-pyruvate which is then converted to IAA by the YUCCA (YUC) family of flavin-containing monooxygenases [113,114,117]. During grape berry development, the gene expression of family members of this two-step pathway suggests that IAA is biosynthesized through the shared action of TAR and YUCCA proteins, and three *TAR* genes and one *YUCCA* gene are highly expressed during development and at the onset of ripening [118]. In the woodland strawberry (*Fragaria vesca* L.) genome, four *TAR* genes have been mapped [119]. In *F. × ananassa*, RNA-seq analysis of the *TAR* analogous genes showed that three of these genes (*FaTAA1*, *FaTAR1*, and *FaTAR2*) were expressed in fruits of this species [31]. This last study showed decreased expression of auxin biosynthesis (*FaTAA1* and *FaTAR2*)-, perception-, signaling (*FaAux/IAAs* and *FaARFs*)-, and transport (*FaPIN*)-related genes at the onset of *F. × ananassa* ripening, from the green to red stages [31]. In the achene, the *TAA1/TAR* genes showed the same pattern, i.e., a sharp decrease at the onset of ripening (green to white stage) and then decreasing to the red stage; the only gene showing high expression was *FaTAR2* which continuously increased its expression level from the green to the red stages [31]. With regard to IAA biosynthesis, only five of the nine *YUCCA* genes mapped in *F. vesca* [119] were expressed in *F. × ananassa* [31]. In the achene, the common pattern was a decrease from green to red stages. In spite of their low expression, some *YUCCA* genes were identified in the red receptacle [31]. Recently, the expression of *TAR* and *YUCCA* genes was

studied using their corresponding promoters fused to a β-glucuronidase (GUS) reporter introduced into *F. vesca* [120]. It was reported that four *FveYUC*s and two *FveTAR*s were expressed mainly in the endosperm and embryo inside the achenes [120]. All these reports suggest the importance of auxin biosynthesis during development and ripening of achenes and receptacle in strawberry fruit.

During plant development, the polar transport of auxin is carried out by the AUX/LAX and PIN transport proteins [121]. Although the signaling mechanism that regulates this transport, and the variation in polar auxin transport, including *PIN* gene expression during fruit growth have been mainly reported in climacteric fruit [122–124], only a few reports are available for non-climacteric fruits. In grape, the use of radiolabeled IAA showed a basipetal distribution in pericarp cells of putative VvPIN proteins by immunolocalization and transcripts of four putative *VvPIN* genes, which decreased according to grapevine fruitlet abscission progress [125]. These antecedents indicate that auxin homeostasis is required for grape berry fruit set. A similar trend has been suggested for strawberry, where a developmental-specific pattern of the transcript of auxin transporters was reported [31,119,126]. The sequenced genome of *F. vesca* showed 10 *FvPIN* [119] and four *FvAUX/LAX* genes [126], and transcriptional analysis in *F. × ananassa* fruit showed the expression of four *FaPIN* genes (*FaPIN1*, *FaPIN4*, *FaPIN5*, and *FaPIN10*) and four *FaAUX/LAX* (*FaAUX/LAX1*, *FaAUX/LAX2*, *FaAUX/LAX3*, and *FaAUX/LAX4*) [31]. All these antecedents suggest a decrease of IAA transport at the onset and during fruit ripening in achene and the receptacle of strawberry. It is notable that most of the transport genes, except *FaPIN10* and *FaAUX/LAX4*, showed higher levels of transcripts in the receptacle compared to achene [31].

IAA conjugation to amino acids by IAA-amido synthetase (GH3) is an important part of auxin homeostasis [127,128], and yet little is understood regarding IAA conjugation in many fruits. Free IAA is the biologically active form of the hormone, with amino acid conjugation leading to IAA inactivation, by storage or degradation [127,129–131]. Therefore, modified forms of IAA are important during degradation, IAA storage or inhibition of auxin signaling pathways [131]. In climacteric and non-climacteric fruit, such as peach and grape respectively, the regulation of the pool of IAA conjugates involves auxin inactivation by conjugation and auxin activation by conjugation hydrolysis [50,132]. In peach, the expression of a putative IAA amidohydrolase like IAA-LEUCINE RESISTANT 1 (ILR1) of Arabidopsis seemed to be counterbalanced by that of a GH3 protein, since both genes were strongly induced by NAA and expressed almost exclusively during fruit ripening [132]. Böttcher et al. (2011) [103] reported that the VvGH3.1 enzyme is involved in berry ripening, and exogenous auxin and ABA application increased the expression of its encoding gene [57], indicating that IAA inactivation by conjugation exists during grape ripening [50,57,120]. In addition to GH3, low expression of putative genes encoding for indole-3-acetate beta-glucosyltransferase (IAGLU), involved in auxin conjugation, and auxin-amidohydrolase involved in auxin activation by conjugation hydrolysis, has been observed during grape ripening [50]. While the expression of the putative gene related to auxin activation by conjugation hydrolysis IAA-amino acid hydrolase 1 (ILR1) was upregulated [50]. In the case of strawberry, transcriptomic analysis during fruit ripening showed that the expression of six *GH3* members was higher in achene, showing highest expression of *FaGH3.1* in the receptacle, which dramatically decreased from green to red stages [31]. Recently, Tang et al. (2018) [133] reported that the only demonstrable IAA amino acid conjugates in the achene of *F. vesca*, and other *Fragaria* species, were aspartate and glutamate, which could act as free IAA reservoirs for growing seedlings. Recently, we have observed in raspberry that before the onset of fruit ripening, fruit size and weight increased along with the expression levels of the IAA-amido synthetase *RiGH3.1* transcript. In addition, the *RiGH3.1* expression was upregulated during in vitro IAA treatment [53]. In turn, the *RiGH3.5* transcript was expressed primarily in flowers and its transcript levels were not significantly affected by IAA treatment [53]. Particularly, the RiGH3.1 amino acid deduced sequence presented the binding motifs for binding of IAA and aspartic acid [53]. The presence of both motifs suggests potential IAA degradation during raspberry ripening.

Other studies related to the role of IAA in the development of non-climacteric fruit has been derived from the analysis of transcription factors that regulate the auxin response. A recent study in cucumber (*Cucumis sativus* L.) has identified two F-box auxin receptors (*TIR1-like* genes) encoding genes *CsTIR1* and *CsAFB2* [134]. These two genes were highly expressed in leaf, female flower and young fruit tissues of cucumber. Furthermore, tomato lines that overexpressed *CsTIR1-* and *CsAFB2*-genes exhibited parthenocarpic fruit compared with the wild-type plants [134]. In grape, the gene encoding for cell elongation bHLH protein (*VvCEB1*) has been suggested as the transcription factor that controls the cell expansion in grape [135]. *VvCEB1* was reported to be induced by auxin and in turn, this gene induced the expression of auxin-related genes in transgenic grape embryos [135]. In *F. chiloensis* fruit, expression analysis for the encoding genes of the transcription factors auxin response factors (ARFs) showed a decrease and increase for *ARF1* and *ARF4* as ripening progressed, respectively [136]. Thus, these reports suggest that not all ARF encoding genes match with the IAA level pattern during strawberry fruit ripening.

The molecular mechanisms for GH3 activity regulation are unknown in fruits. Even so, in Arabidopsis, the auxin response factor ARF17 has been described as a negative regulator, and ARF6 and ARF8 as positive regulators for *GH3.3-*, *GH3.5-* and *GH3.6*-expression during adventitious root initiation [137]. Therefore, the regulation of *GH3* expression by ARFs or other transcription factors (e.g., AREB/ABFs, MADS-box, ERFs) and its effect on enzymatic activity during fruit ripening should be evaluated.

Transcriptional regulators such as the *MADS-box* genes are involved in the development and ripening of both climacteric and non-climacteric fruit [55,138–140], indicating a possible conserved role of these regulators in fruit ripening [1]. The putative MADS-box protein RIPENING INHIBITOR (RIN) is an auxin response transcription factor involved in the ethylene-regulated gene expression [140]. It has been reported that in tomato fruit, the RIN MADS-box transcription factor could bind to the promoter of the cell wall hydrolases encoding genes, as well to the promoter of ACS encoding genes [141]. Recently, it has been demonstrated that RIN is not necessary for the initial induction of tomato ripening, suggesting that the expression of ripening-related genes in the RIN knock-out mutants could be explained by the activity of a RIN homologous, where the RIN action could be substituted by other MADS-box protein complexes that bind to RIN recognition sites [142,143].

*2.3. Gibberellins*

Gibberellins (GAs) are tetracyclic diterpenoid compounds involved in developmental processes such as seed germination, cell division and elongation, flower induction and development, and fruit growth [144–146]. Even though hundreds of different GAs have been described in plants, only a limited number of these molecules were found to be biologically active [146,147]. Their homeostasis was determined by the rate of bioactive GA biosynthesis, including gibberellin 3-oxidase (GA3ox), and inactivation by hydroxylation reactions by the action of gibberellin 2-oxidase (GA2ox) [146–148]. The first report on exogenous GA application and its effects on strawberry receptacle development were reported by Thompson in 1969 [149]. Afterwards, many other studies have complemented GA association with fruit ripening, especially in strawberry fruit [146,150,151]. Bioactive GA1, GA3, and GA4 were reported during strawberry fruit development. High levels of GA4 were reported in the receptacle tissue, with a peak level at the white stage of strawberry fruit development. In addition, the expression of genes encoding for GA pathway components, including the receptors, such as the FaGID1 (GIBBERELLIN-INSENSITIVE DWARF1b), and FaGID1c, the DELLA FaRGA (REPRESSOR OF GA) and FaGAI (GA-INSENSITIVE), and enzymes related to GA biosynthesis such as gibberellin n-oxidase (FaGA3ox), and GA hydrolysis (FaGA2ox) were also reported at high levels in the receptacle tissue during strawberry fruit development [146]. FaGID1c bound GA in vitro, interacted with FaRGA, both in vitro and in vivo, and increased GA responses when ectopically expressed in Arabidopsis [146]. These antecedents suggest a critical role for GA in the development of the strawberry receptacle [146]. The overexpression of the Gibberellin Stimulated Transcript 2 (*FaGAST2*) gene in different strawberry

transgenic lines produced a reduction in fruit size, showing that the parenchymal cells were smaller than observed in the control fruits, indicating a relationship between the expression of *FaGAST2*, strawberry fruit cell elongation and the subsequent increase in fruit size [152]. Silencing of *FaGAST2*, through RNA interference methods, showed an increase in *FaGAST1* expression, without changes in fruit cell size, suggesting that both transcription factors were involved in fruit development and ripening, determining the strawberry fruit cell size [152].

Exogenous GA application and its role in viticulture have been studied by many researchers [153–161]. The exogenous pre-bloom application of gibberellic acid ($GA_3$) to grape cultivars, that is normally used to establish early ripening [153], also induced seedlessness [154,158], and enhanced berry size in seedless cultivars [155–157]. Transcriptome sequencing conducted to identify microRNAs during $GA_3$ application suggests that grapevine miRNAs (Vvi-miRNAs) might have a role in grape berry development and response to environmental conditions [159]. Application of $GA_3$ to inflorescences at the pre-bloom stage stimulated flower opening and promoted fruit coloring along with seed abortion in 'Kyoho' grape [158]. Moreover, GA application produced changes in the levels of transcripts related to reproduction, cellular processes, hormones, and secondary metabolism, including the scavenging and detoxification of reactive oxygen species [158].

The role of GAs in fruit development and ripening of the *Rubus* genus has not been studied extensively. It has been reported that the external application of some GAs induced parthenocarpic fruit development in cloudberry (*Rubus chamaemorus* L.), a northern wild berry [162]; and external application of prohexadione–calcium (ProCa), a gibberellin biosynthesis inhibitor, reduced the number of flowers in raspberry [163].

## 2.4. Ethylene

The role of ethylene during non-climacteric fruit ripening has attained increasing attention among researchers in recent years. It has been reported that ethylene plays a major role in the fruit ripening of different grape varieties, with an ethylene peak occurring before véraison [56,72]. Treatment of fruits with a specific inhibitor of ethylene receptors, i.e., 1-methylcyclopropene (1-MCP), reduced berry diameter and ripening-related parameters, such as anthocyanin accumulation in 'Cabernet Sauvignon' berries [56]. Increased berry size associated with high levels of transcripts related to aquaporins, polygalacturonases, xyloglucan endotransglucosylases (XTHs), cellulose synthases, and expansins were reported as a result of ethylene application [164]. Also, 1-MCP application prior to véraison reduced ABA content in 'Muscat Hamburg' grape [71], suggesting an association between ethylene and ABA during ripening. Despite the low concentration of ethylene in *F. × ananassa*, its production was sensitive at various fruit stages, being moderately high in green fruit, with lower levels in white fruit, and an increment at red stages of fruit maturation [165]. The increased ethylene production at the red fruit stage was accompanied by an improved respiration rate and resembled that of climacteric fruit [165]. Although raspberry fruit belongs to the non-climacteric fruit category [9,166,167], ethylene production and increased respiration rate were detected at the white fruit stage and continued to increase until full maturity [3,10,53]. Therefore, this climacteric pattern of ethylene production in raspberry fruit (i.e., showing a constant increase during ripening) is very different from other non-climacteric fruit such as strawberry [165] and grape [56,71]. In fact, strawberry [165] and grape [56,71] show a peak in ethylene activity during early fruit development, which is a typical pattern in non-climacteric fruits. The ethylene production of raspberry fruit was inversely related to firmness loss, and the receptacle has been described as the main ethylene source [3,10,53,168]. In addition, raspberry fruit (drupelets still attached to their receptacle) at the white stage showed a delay in firmness loss during in vitro treatment with 1-MCP during storage at 10 °C [10]. These results suggest that softening during raspberry ripening could be regulated partially by ethylene.

The expression of genes encoding for the ethylene biosynthesis-related enzymes 1-aminocyclopropane- 1-carboxylic acid synthase (ACS) and 1-aminocyclopropane-1-carboxylic acid oxidase (ACO) has been reported in non-climacteric fruit. In strawberry, four *FaACS* genes [169] and three *FaACO* genes have been isolated [26,169] with different expression patterns during ripening. In different cultivars of grape, i.e., 'Cabernet Sauvignon', 'Muscat Hamburg' and 'Thompson Seedless', the expression of *VvACO* genes has been found in association with the ethylene peak, just before véraison [72,164,170]. In strawberry, the expression of two *FaACO* and ethylene response sensor (*FaErs1*) genes has been reported during fruit development, and also, in fruits treated with ethylene and the auxin analogue 1-naphthaleneacetic acid (NAA), a close relationship between the expression of these genes and ethylene production was observed [26]. In raspberry, *RiACS1* and *RiACO1* genes were expressed more at the red and overripe stages [169]. Interestingly, levels of both transcripts were more than three-fold higher in the receptacle tissue than in the drupelet at the overripe fruit stage. The expression of *RiACS1* in raspberry was similar to that observed for *FaACS1* and *FaACS2* genes, highly expressed in green and white strawberry receptacles [169]. The deduced amino acid sequence RiACO1 was homologous with the FaACO1 protein sequence (73.4%), whose transcript was detected in strawberry receptacles [169]. In addition, RiACO1 was homologous with SlACO4 (79.3%), whose transcript was expressed during ripening of tomato fruit [171]. These antecedents suggest that a similar mechanism could regulate ethylene biosynthesis in climacteric and non-climacteric fruits.

During the ripening of climacteric and non-climacteric fruits, several studies have reported expression changes of the genes encoding for the ethylene signaling cascade-related components [172–174]. These genes include those encoding for the ethylene receptor (ETR), ethylene response sensor (ERS), ethylene insensitive (EIN), and constitutive triple response (CTR1), the second element in ethylene signal transduction at the endoplasmic reticulum (ER) membrane level, acting as a communicator between ETRs and EIN2s [172–174]. The ethylene receptor (ETR) family belongs to a family of transmembrane proteins that are found in the ER and that bind ethylene, forming a stabilized dimer with two disulfide bonds at the N-terminus. The ETRs are the first elements of ethylene signaling, acting as negative regulators of the signaling cascade, blocking downstream signal transduction in the absence of ethylene [172–174]. Therefore, downregulation of *SlETR4* produced early ripening of transgenic tomatoes [175,176].

During the development of the grape berry, *VvETR2* showed a high expression level at the onset of ripening, while *VvETR1* expression remained constant, and *VvERS1* and *VvEIN4* had greater expression levels at first days of post-anthesis [174]. In strawberry, the expression of three genes encoding for the receptors *FaEtr1*, *FaErs1* and *FaEtr2* have been observed concomitant with increased synthesis of ethylene. The observation that *FaEtr2* was mostly expressed in ripe strawberry fruits, suggests that even the little ethylene produced by ripening strawberries could activate ripening-related physiological process [26]. In addition, dataset analysis of the expression pattern of the ethylene biosynthesis-related gene S-adenosyl-l-methionine synthase *FaSAMS1*, and the signaling-related gene *FaCTR1*, showed that the transcription induction of both genes was coincident with increase in ethylene production the during ripe fruit-coloring [176]. It was assumed that *FaSAMS1* and *FaCTR1* might have a major role in the evolution of attributes during fruit ripening [160]. Downregulation of both transcripts via the tobacco rattle virus-induced gene-silencing (VIGS) system not only repressed red coloration of fruit and firmness changes, but it also induced ethylene production, affecting ethylene-signaling components [176]. Importantly, the application of ethephon (a synthetic ethylene releaser) stimulated natural strawberry fruit softening and red coloration in these transgenic fruits by the partial rescue of anthocyanin biosynthesis without significantly affecting fruit firmness. This may suggest that FaCTR1 positively controls strawberry fruit ripening [176], but it still remains to be clarified if ethylene has a potential role in stages prior to ripening in this non-climacteric fruit.

### 2.5. Jasmonates

Jasmonic acid (JA) and its bioactive isoleucine conjugate (JA-Ile) are among the most important signals for both biotic and abiotic stress responses in plants and are active in root growth, seed germination, stamen development or senescence [177]. Recently, Garrido-Bigotes et al. (2018) [51] observed that a synchronized downregulation of the JAs endogenous levels, including JA-Ile, and their biosynthetic genes takes place from flowering to ripening stages of strawberry fruit [51]. Also, higher levels of JA and JA-Ile have been reported in early developing grape berries followed by a sudden reduction during progression to ripe stages [178]. Thus, JA-Ile could be associated with early fruit development in both strawberry and grape berry. It was reported that proanthocyanidin (PA) accumulation in developing strawberry and grape berry showed a similar pattern to JA-Ile, occurring early in fruit development and decreasing as the fruit ripens [179,180]. We observed an increase in PA content by the application of a chemical inhibitor for the key JA-Ile biosynthetic enzyme JAR1, suggesting that the JA pathway could be related to PAs biosynthesis in strawberry fruit [181].

In *F. chiloensis* fruit, exogenous application of the methyl ester of JA (methyl jasmonate, MeJA) changed the expression profiles of ripening-related genes including those related to ethylene and JA biosynthesis [182]. Moreover, MeJA application increased anthocyanin accumulation, since MeJA upregulated the expression of genes related to the anthocyanin biosynthesis pathway, as well as chalcone synthase (*FcCHS*), chalcone-flavonone isomerase (*FcCHI*), flavanone 3-hydroxylase (*FcF3H*), dihydroflavonol 4-reductase (*FcDFR*), anthocyanin synthase (*FcANS*), and anthocyanidin 3-*O*-glucosyltransferase (*FcUFGT*). Associated with the higher amounts of anthocyanins, MeJA also induced the expression of JA biosynthetic genes in *F. chiloensis* fruits, i.e., 13-lipoxygenase (*FcLOX*), allene oxidase synthase (*FcAOS*) and 12-oxophytodienoate reductase 3 (*FcOPR3*) [182]. Alternatively, the exogenous application of MeJA to *F. × ananassa* fruit generated a higher anthocyanin increase along with the accumulation in JA, JA-Ile and MeJA levels [51]. In grape cell suspensions JA increases anthocyanin production [183] and MeJA application increased PA content in two wine grape varieties [184]. In raspberry, one of the aspects more studied is the molecular and biochemical factors that determine color. It was reported that jasmonates enhanced PAL activity and promoted a significant increase in polyphenol compounds such as ellagic acid, quercetin, and myricetin [185,186].

The biological processes mediated by JA-Ile need the activation of the JA signaling pathway, in which the F-box CORONATINE INSENSITIVE1 protein (COI1) associated with JASMONATE ZIM-DOMAIN (JAZ) form the JA-Ile co-receptor [187–189]. If JA-Ile levels are low, JAZ repressors bind to MYC2 and other transcription factors suppressing the expression of early JA-responsive genes, whereas if JA-Ile level rises, COI1 binds to JAZs that are subsequently degraded by the 26S proteasome after ubiquitination by ubiquitin ligase complex SCF$^{COI1}$ [190]. In grape, 11 JAZ members have been described which respond to biotic and abiotic stresses and hormone treatments [191]. We recently demonstrated that 12 putative strawberry JAZ proteins and two MYC TFs encoding genes exhibited high expression in flowers and at early fruit stages of strawberry that matched with the downregulated pattern of JA-Ile observed during the development of this fruit [51].

### 2.6. Brassinosteroids

Brassinosteroids (BRs) are polyhydroxy steroid phytohormones that play critical roles in cell division and elongation, vascular differentiation, flowering, pollen development and photomorphogenesis [192]. BRs have also been reported to be involved in development and ripening of tomato, cucumber, grape, and strawberry fruits [49,193–195]. In turn, it has been observed that Arabidopsis mutants deficient in BR biosynthesis or perception, including dwarf plants, presented numerous deficiencies in developmental pathways that could not be rescued by BR treatment [196]. In grape berries, the application of BRs (as epibrassinolide, BL) increased berry colouring and promoted ripening. Conversely, the application of a BR biosynthesis inhibitor, i.e., brassinazole (BZ), showed a contrary effect [194].

BR 6-oxidase is the enzyme that catalyzes the conversion of 6-deoxocastasterone to the bioactive BR castasterone [194]. It has been reported that the over-expression of *V. vinifera* BR 6-oxidase gene (*VvBR6OX1*) restored the phenotype of the dwarf tomato plant (that lacks a functional endogenous DWARF gene), reaching a normal height similar to a wild type plant [194]. In strawberry fruit, BL treatment promoted fruit ripening and the accumulation of FaBRI1 receptor transcript [49,197]. Moreover, transient silencing of the *FaBRI1* gene caused a delay in fruit ripening, so that the receptacle remains white [49]. These antecedents show that BR signaling could play a central role in non-climacteric fruit ripening. BRs have been proposed to be the first signal for ripening of the grape berry, possibly through modulation of ethylene content [104]. With regard to genes that respond to BRs, the gene encoding for the late embryogenesis abundant (LEA) domain protein (LDP1), has been expressed in early developmental stages of *F. chiloensis* and *F. vesca*, particularly in receptacles [198]. The promoter region of *LDP1* gene possesses multiple BRs and ABA response motifs. It has been reported that the transiently expressed *FcLDP1* promoter-GFP fusions was regulated by BRs and ABA, suggesting that these two hormones regulate *FcLDP1* expression during *F. chiloensis* fruit development [198].

*2.7. Cytokinins*

The role of cytokinins (CKs) in non-climacteric fruit development and ripening is a topic that needs further investigation. Bombarely et al. (2010) [199] described two genes belonging to the CK signal transduction pathway from fruit cDNA libraries of several varieties of *F. × ananassa*, identifying a histidine phosphotransferase protein (AHP) and a nuclear response regulator (ARR) gene. Some years after, Kang et al. (2013) [119] compared the *F. vesca* transcriptomes of pre- and post-fertilization stages during fruit development and observed 17 differentially expressed genes (DEGs) related to CK biosynthesis, signaling, and degradation in four different fruit tissues. It has been reported that CKs are important in the initial stages of strawberry fruit development, similar to that of the role described for CKs in the early fruit development of climacteric fruit, such as tomato [200]. In grape, the synthetic cytokinin forchlorfenuron [*N*-(2-chloro-4-pyridyl)-*N*′-phenylurea], known as CPPU, was reported to be involved in the increase of berry weight, but a decrease in sugar and anthocyanin content was observed [201]. The only report describing the role of CKs in the *Rubus* genus suggests that GAs could act synergistically with CKs during flower induction in raspberries [202].

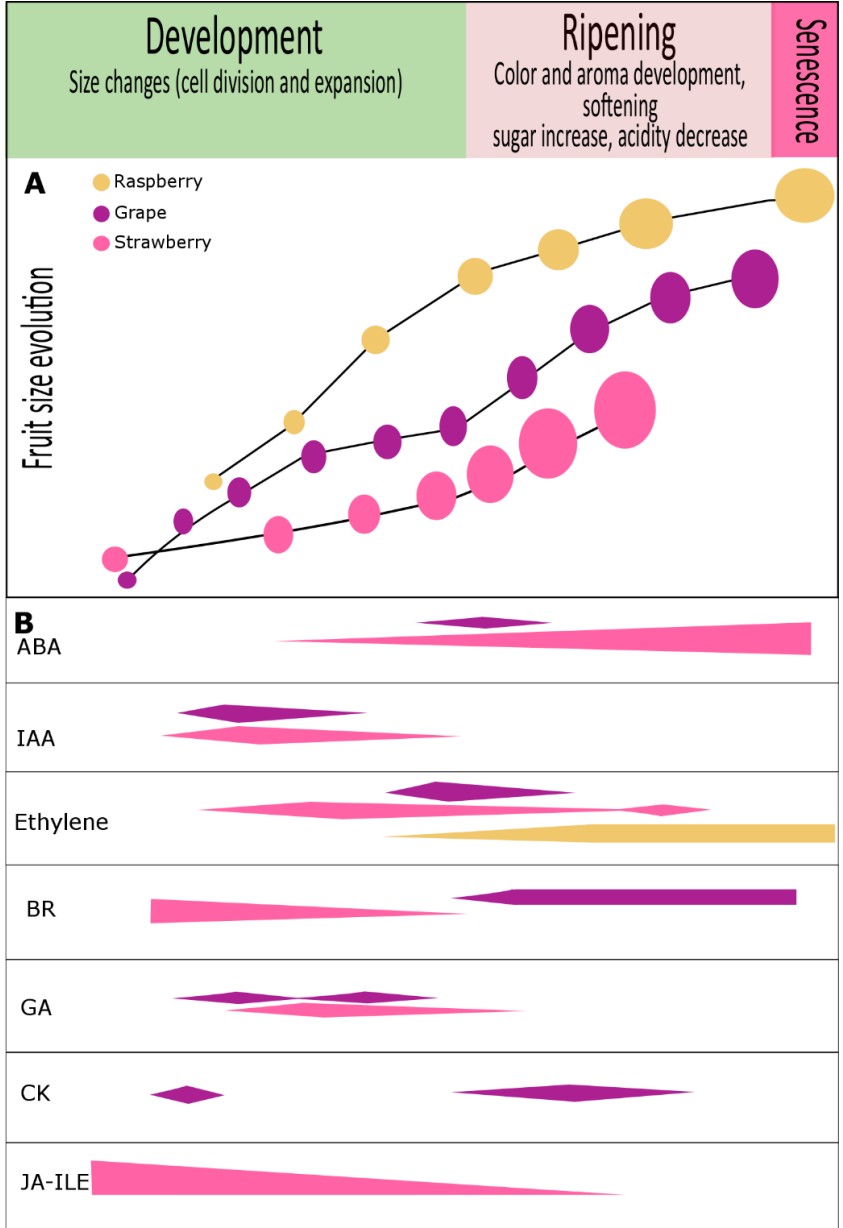

**Figure 1.** Phytohormone content evolution during development and ripening of grape, strawberry and raspberry fruits. (**A**) Developmental trends according to fruit size in grape (purple) [203], strawberry (pink) [18,51] and raspberry (yellow ochre) [53]. (**B**) Variations in hormone levels are shown for abscisic acid (ABA), auxin (IAA), ethylene, brassinosteroids (BR), gibberellins (GA), cytokinins (CK), and jasmonic acid isoleucine conjugate (JA-Ile). In grape, numerous reports have revealed an important role for auxins, cytokinins, and gibberellins ($GA_3$) associated with cell division and fruit set. At véraison, a noticeable increase in abscisic acid levels was reported to be crucial for ripening progression, but during ripening a single and constant hormonal pattern is not the rule [203]. In strawberry, abscisic acid is thought to be important, but the roles of other hormones including IAA, BR, $GA_1$, $GA_3$, $GA_4$, JA-Ile, and ethylene, were also suggested to be involved in fruit development. Although raspberry fruit has been described as non-climacteric fruit, ethylene can be detected at the onset of ripening and continues to increase until full ripening [3,10,53]. There are no reports regarding other hormones in raspberry, although we could detect the presence of auxin in green stages and our transcriptome analysis has shown differential expression of phytohormone-related genes during fruit development and ripening [204]. More details in the text.

## 3. Hormone Crosstalk

A single hormone can regulate various processes, and at the same time, multiple hormones could impact a single process as well [205,206]. Some pieces of evidence for auxin–ethylene crosstalk was reported in *Capsicum*, and the expression of the *CsGH3* gene was upregulated by ethylene application [207]. Similarly, the upregulation of the expression of GH3 family-related genes by ABA and ethylene application during fruit ripening in grape and by other phytohormones in tomato suggests that auxin can crosstalk with ethylene, ABA, and other phytohormones [2,57,208,209]. ABA promoted fruit ripening by affecting ethylene biosynthesis in fleshy fruits [210,211]. For example, the exogenous application of ABA increased the expression of ethylene biosynthesis-related genes, such as *ACS2*, *ACS4*, and *ACO1*, in tomato. Conversely, exogenous treatment with the ABA inhibitor fluridone showed a downregulation of these genes, suggesting that ABA could regulate the ethylene biosynthetic pathway and vice versa in climacteric fruit such as tomato [2,212]. Contrary to that of climacteric fruit such as tomatoes, where maximum ABA levels were observed preceding ethylene production [213,214], ethylene levels in grape remained low at the onset of ripening [56].

Data from several transcriptomic analyses, including our data on raspberry [204], suggest a coordinated role of different phytohormones during fruit development and ripening in non-climacteric fruit such as grape [104,215] and strawberry [31,216]. However, an in-depth analysis is still required to fully elucidate the role of phytohormone crosstalk in non-climacteric fruit ripening. When compared with climacteric fruit, much less information is available on phytohormone crosstalk in non-climacteric fruit. In the following sections, we briefly summarize the available literature on plant hormone crosstalk during fruit development of grape, strawberry and raspberry fruit (Figure 2).

### 3.1. Grape

Transcriptional analysis of grape berries after application of naphthalene acetic acid (NAA) one week before véraison showed that ABA biosynthesis- and perception-associated genes were downregulated, while ethylene biosynthesis-associated ones were induced [104]. In addition, interaction between ethylene and auxin has been suggested as a mechanism of grape ripening control [118]. The expression of several *TAR* genes that encodes for the enzyme that converts tryptophan to indole-3-pyruvate, the first step of IAA biosynthesis, was induced by the application of ethephon [118]. The induction of *TAR* genes was complemented by increased IAA and IAA-Asp concentrations, suggesting that elevated concentrations of ethylene at the onset of ripening might lead to increased production of IAA in grape berry ripening [118].

The developmental phase in many fruits was reported to be controlled by a hormonal balance between GA and auxin [203]. Auxin induces the generation of parthenocarpy. Treatment with GA at the pre-bloom stage of berries showed an upregulation of the GA signaling gene *VvDELLA*, together with a decrease of the expression of the genes encoding for negative regulators of fruit set initiation, i.e., AUX-IAA protein, VvIAA9 and auxin response factor (ARF) VvARF7. Also, the upregulation of *VvGH3.2* and *VvGH3.3* expression, without significant effects on *VvYUC2* and *VvYUC6* expression, was reported. This suggests that GA signaling is associated with IAA signaling via VvDELLA during parthenocarpy in grape [160]. Exogenous application of the auxin analogue 4-chlorophenoxyacetic acid (4-CPA) to ovaries of 'Fenghou' grape, was reported to promote fruit set, depending on subsequent biosynthesis of gibberellin $GA_3$ [217].

Calcium ($Ca^{2+}$) has many roles in plants serving as a second messenger, and in cell wall polysaccharide interactions, crucial in stress responses, cell wall growth, and remodeling, and plant tissue development [218–220]. However, its molecular implication on phytohormone crosstalk regulation is still not very clear. Studies of ABA and MeJA applied alone or in combination with calcium to grape suggested a calcium-hormone interplay that regulated the expression and activity of flavonoid biosynthetic enzymes [221].

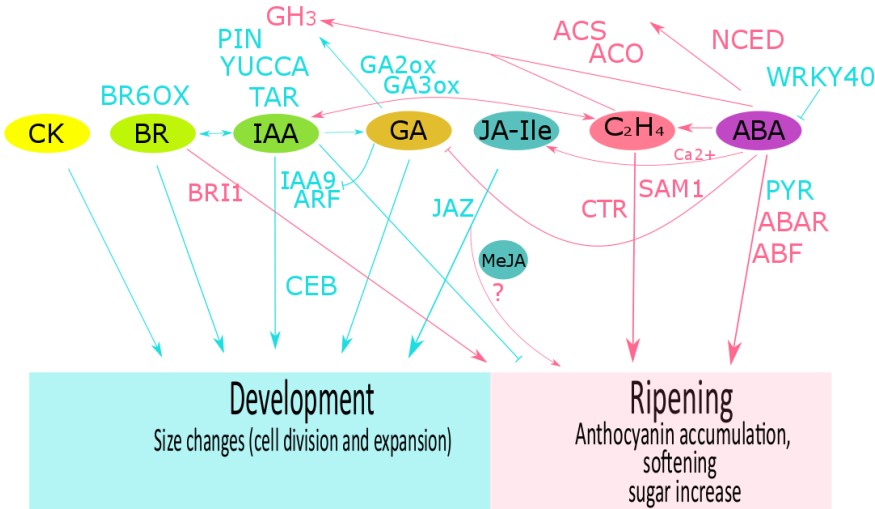

**Figure 2.** Phytohormone crosstalk described in grape, strawberry and raspberry fruits. The role of hormones and their related components during fruit development (cyan color) and during ripening (pink color) are shown for abscisic acid (ABA), auxin (IAA), ethylene ($C_2H_4$), brassinosteroids (BR), gibberellins (GA), cytokinins (CK), and jasmonic acid isoleucine conjugate (JA-Ile). In many fruits, development is controlled by a balance between GA and auxin. In grape, GA downregulates *VvIAA9* and *VvARF7* expression and then auxin transport, and upregulates *VvGH3* expression. In turn, auxins could promote fruit set and biosynthesis of gibberellins. IAA and BRs crosstalk has been suggesting as an important mechanism to control fruit growth. During grape ripening, an interaction between ethylene and auxin has been described, and calcium-ABA-JA interplay could regulate the expression of flavonoid biosynthesis. In strawberry, ABA, auxin and GA crosstalk regulate the transition of development to ripening. Methyl jasmonate (MeJA) application at the onset of ripening stimulates ethylene and anthocyanin biosynthesis. However, the JA-Ile content has been suggested as important for fruit development. *FaNCED1* downregulation by MeJA application suggests an antagonistic role of JA on ABA biosynthesis. In raspberry, exogenous IAA and MeJA regulate the anthocyanin content. TAR: tryptophan aminotransferase; YUCCA: flavin-containing monooxygenases; PIN: auxin transporter (PIN-FORMED protein); GH3: IAA-amido synthetase; ARF: auxin response factor; CEB1: transcriptional factor cell elongation bHLH protein; NCED: 9-*cis*-epoxycarotenoid dioxygenase; PYR1: ABA receptor (pyrabactin resistant); FaABAR: ABA receptor; ABF2: ABA transcriptional factor; GA-n-ox: gibberellin n-oxidase; ACS: 1-aminocyclopropane-1-carboxylic acid synthase; ACO: 1-aminocyclopropane-1-carboxylic acid oxidase; SAMS1: S'adenosyl-l-methionine synthase; ETR: ethylene receptor; CTR1: constitutive triple response gene; ERS1: ethylene response sensor; EIN: ethylene insensitive; JAZ: jasmonate zim-domain protein; BR6OX: BR 6-oxidase; BRI1:BR receptor. More details in the text.

IAA and BRs have been described as key regulators that determine the growth of several plants, influencing cell division and elongation in various developmental contexts [222]. Despite the reports available about the role of BRs in fruit quality, their crosstalk with other phytohormones can only be suggested based on transcriptomic analyses. Comparison of transcriptomic analyses between varieties of small and large grape berries have shown that differentially expressed transcripts were mainly related to auxin, ABA, ethylene, but also to BRs and gibberellins [215]. The comparison between small and large grape berries during fruit development revealed significant differences in softening rate, firmness, and sugar accumulation. The transcriptional dataset showed DEGs related to auxin, ABA and the ethylene hormone pathway between varieties of small and large grape berries [215]. The same study showed the specific regulatory motifs related to bZIP, bHLH, AP2/ERF, NAC, MYB, and MADS-box transcription factors in the cis-regulatory element of the promoter regions of DEGs related to fruit texture, aroma, and flavor, and hormones, observing differences in the promoter regions between small and large grape berries [215].

Exogenous application of GA$_3$ and IAA to grape reduced polar auxin transport, but only GA$_3$ treatment decreased VvPIN transcript abundance. The GA biosynthesis blocking allowed increased IAA polar auxin transport, suggesting that polar auxin transport depends on GA content [215]. In tomato, molecular studies revealed that an auxin response factor (*SlARF7*) was involved in the regulation of the crosstalk between auxin and GA, and silencing of this gene caused parthenocarpic fruit formation due to augmented auxin and GA responses, and further suggest that *ARF7* acts as a changer of the GA response in early fruit development as has been reported in *Arabidopsis* and tomato [61,223]. However, the potential role of ARF as GA-response regulator during grape development still needs to be investigated.

## 3.2. Strawberry

In strawberry fruit, expression analysis showed that auxin and ABA are the main hormones responsible for the transition from development to ripening stages [224]. Auxins are responsible for the development of the receptacle and, at the same time, prevent ripening by repressing key ripening-related genes. ABA regulates the expression of the majority of genes involved in ripening, while ethylene and gibberellins do not seem to play a noticeable role during maturation [224]. After harvest of strawberry, exogenous IAA delayed strawberry ripening, while exogenous ABA treatment had the contrary effect. However, the combined treatment did not show any of these effects on the postharvest ripening of the strawberry fruits [225]. A comparison of transcriptomes of fruit under the individual or combined treatments revealed that there were differentially-expressed unigenes in response to the ABA and IAA combination treatment [225]. Exogenous IAA application upregulated IAA signaling-related genes such as *AUX/IAA*s and *ARF*s, and downregulated cell wall degradation-, sucrose and anthocyanin biosynthesis-related genes. Conversely, ABA induced the expression of genes related to fruit softening and signaling pathways as that encoding for the S-phase kinase-associated protein (SKP1) component of the Skp1-Cullin1-F-box (SCF) complex that facilitates ubiquitin-mediated protein degradation [224]. The expression of a C-type MADS-box gene in strawberry [*SHATTERPROOF-like* (*FaSHP*)] was downregulated by auxin and upregulated by ABA [55]. In addition, its promoter presented responsive elements to auxin and ABA, explaining how *FaSHP* could be controlled by these two hormones [55].

Regarding the possible role of calcium in hormone crosstalk in strawberry fruit, the application of calcium in combination with auxin (as NAA) reduced the transcript level of the cell wall modifying-related genes *FcPG1*, *FcPL* and *FcEG*1 (encoding for polygalacturonase, pectate lyase and endoglucanase, respectively) [226]. These results suggest that the auxin repressor role in strawberry ripening could be reinforce by calcium, although more research is needed to clarify this interaction.

Recently, interactions and regulation between auxin, GA and ABA during fruit development and ripening of *F. vesca* fruit have been reported [227]. This study reported that the increase in the GA content at the early stages could antagonize the inhibitory effect of ABA, whereas the quick drop of ABA during early stages of fruit development guarantees fruit growth induction by GA [227]. The ABA catabolism gene *FveCYP707A4a* was reported as an important crosstalk point for auxin, GA and ABA, regulating the transition from the early growth phase to ripening phase [227].

MeJA application at the onset of ripening, especially at the white stage, stimulated ethylene biosynthesis by an increase in ACO activity [228]. In *F. chiloensis* fruit, the application of MeJA increased the expression of both *ACO* and *ACS* genes suggesting that several ripening effects triggered by exogenous MeJA could be mediated by ethylene [182]. MeJA treatment of climacteric fruit such as apple enhanced the expression of *MdMYC2*, a gene encoding a transcription factor involved in the JA signaling pathway. The MdMYC2 directly bound to the promoters of *MdACS1* and *MdACO1* genes, enhanced their transcription, and then increased ethylene biosynthesis [229]. However, the potential role of the MYC2 transcription factor on ethylene and MeJA crosstalk must still be elucidated in strawberry. With regard to JA-ABA crosstalk in strawberry, the JA pathway could act antagonistically with ABA for anthocyanin accumulation in strawberry fruit. We reported an increase in anthocyanin content, with an associated decrease in ABA levels, after MeJA treatment, going along with *FaNCED1* downregulation, suggesting an antagonistic association from the JA to the ABA pathway in strawberry [51].

*3.3. Raspberry*

With regard to hormonal crosstalk in raspberry, recent studies suggested that anthocyanin metabolism is regulated by the interplay of multiple hormonal signs, including IAA and MeJA [186]. This study showed that IAA downregulated and MeJA clearly upregulated the expression of the genes encoding for transcription factor MYB10 and anthocyanin synthase (ANS), two genes related to anthocyanin biosynthesis, suggesting an opposite effect of both hormones in the regulation of anthocyanin accumulation in fruits [186]. Although there is still a lot to study in terms of raspberry fruit ripening regulation, our de-novo assembly analysis during different developmental stages showed that the DEGs included transcripts related to synthesis and response to ABA, enzymes with GH3 activity, transmembrane transporter for influx and efflux of auxin, proteins related to response to auxin, proteins related to BR biosynthesis and response, and proteins related to ethylene biosynthesis and perception [204]. All this information suggests a role for different hormones during raspberry development. In our experiments we have observed an increase of ethylene during the IAA treatment of raspberry [45]; however, the possibility of a stress effect of the treatment should not be ruled out.

The loss of membrane integrity during the decline in fruit quality has been associated with 1-phospholipase D (PLD), a phospholipid-degrading enzyme involved in initiating membrane catabolic events that is highly active in berries such as strawberry and raspberry [230,231]. The pre-harvest application of aqueous spray containing hexanal (HC), an enhancer of fruit shelf-life, during fruit development showed significant downregulation of transcript levels of three PLD encoding genes and its associated enzymatic activity, as well as upregulation of the expression of the genes related to calcium-binding protein such as annexin and calmodulin-binding transcription activators [231]. All these changes were associated with the significant increase of the pulling force necessary to detach the berry from the receptacle and abnormal calcium crystalline depositions on the epidermal drupelet [231]. These antecedents suggest a potential crosstalk between hexanal, phospholipase D activity and calcium in delaying fruit softening and in prolonging the storage life of raspberry. However, the relation of these effect with phytohormones still needs to be deciphered.

## 4. Conclusions

In conclusion, the molecular dissection of non-climacteric fruit models (i.e., grape, strawberry, and other less-studied species such as raspberry) has shown that various phytohormones (ABA, auxin, ethylene, and others) act together or regulate each other to affect several molecular and biochemical processes that contribute to fruit quality at the onset of ripening. The growing information on the identification and characterization of key genes associated with the signaling and perception of these hormones in grape and strawberry could impact and widen our knowledge of other non-climacteric fruit ripening process, including those much less studied such as *Rubus* fruits species, and may help improve design strategies that could improve postharvest quality and food security.

**Author Contributions:** L.F., C.R.F., and M.V. wrote the paper. All authors read and approved the final manuscript.

**Funding:** Financial support by the National Commission for Scientific and Technological Research (CONICYT, Chile), grants FONDECYT/Iniciación 11110438 to L.F.; FONDECYT/Regular 1181310 to C.R.F; FONDECYT/Regular 1140817 to M.V. and L.F.; CONICYT-Regional GORE Valparaíso R17A10001 to L.F.

**Conflicts of Interest:** The authors declare no conflict of interest. The founding sponsors had no role in the writing of the manuscript, and in the decision to publish the results.

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
