# Peer review of "Recent Advances in Hormonal Regulation and Cross-Talk during Non-Climacteric Fruit Development and Ripening"

_horticulturae, doi:10.3390/horticulturae5020045_

Round 1

Reviewer 1 Report

The manuscript (horticulturae-449187) entitled "Recent advances in hormonal regulation and cross-talk during non-climacteric fruit development and ripening" by Fuentes et al. was reviewed for publication in Horticulturae.

General Comments

The manuscript has the objective to review current knowledge about hormonal cross-talk during non-climacteric fruit developing and ripening. This would be a timely review on an interesting subject, but the manuscript needs considerable revisions. For example, the introduction needs to be reorganized to focus on the main topic, non-climacteric fruit. As the introduction is currently presented, it is not clear from the beginning which fruits are climacteric or non-climacteric. This should be addressed early in the introduction to put into context the main focus of the manuscript. Secondly, what is known about cross-talk in climacteric fruit? At least a brief summary should be included with the most recent articles on the subject referenced. Another important point, the manuscript needs extensive English editing and therefore corrections by and native English speaker is recommended during the revisions. I’ve made some corrections below noted with “grammar”, but there are many corrections to be made throughout the manuscript. In addition, need to provide full names of genes and proteins with abbreviations throughout manuscript. There are also some very long sentences that are hard to follow that need to be made more concise for clarity. Finally, the last section on hormonal cross-talk has many difficult sections to understand. This section needs to be carefully rewritten to improve clarity.

Corrections

Throughout manuscript (Lines 59, 291, 296, 376 and , the use of the word being is incorrect, fo example, Line 59, being the study of genes. Please check and correct throughout manuscript. Need to spell out gene and protein names with abbreviations, and then use abbreviations.

Line 38, “Evolutionarily, have been reported that…” is grammatically incorrect. Please correct.

Line 44-Line 50, are these characteristics shared by both climacteric or non-climacteric fruit?

Line 60-61, raspberry and blackberries are introduced here, but it is not indicated whether they are climacteric or non-climacteric.

Line 64, grape, strawberry and raspberry are mentioned here, but it is not indicated whether they are climacteric or non-climacteric.

Line 67, Glen Clova and Glen Prosen, Chilean strawberry (Fragaria chiloensis) and cultivated strawberry (Fragaria × ananassa), climacteric or non-climacteric?

Line 76, raspberry and strawberries are introduced here, but it is not indicated whether they are climacteric or non-climacteric.

Line 81, the last paragraph in the Introduction and finally there is information about non-climacteric fruit. The title of the review suggests a focus on non-climacteric fruit, so I think the introduction could be improved by focusing on the main characteristics of climacteric and non-climacteric fruit, that will lead up to the main topics of the manuscript. Examples of non-climacteric fruit discussed in the manuscript should be mentioned early in introduction.

Line 85, what about respiration, are there differences between climacteric and non-climacteric fruit?

Line 97, Need to give full names for NOR, TAGLI and RIN with the abbreviations.

Line 103, “normal fruit growth even in the absence of fertilization”, you mean parthenocarpy?

Lines 121-122, what affect, increase gene expression?

Line 123, do you mean genes for L-ascorbic acid and folic acid are also increased? Please clarify.

Line 134, “g Asubstantial” ? Please correct.

Line 138, grammar, have been

Lines 139-140, please give full names of genes/proteins.

Line 141, binding, not biding

Line 145, grammar and spelling, “), activated the expresion", please correct

Line 148, grammar, becomes high

Line 150, grammar, have recently

Line 151, write out full name gene/protein name, VlPYL1

Line 156, also improved

Lines 171-172, “accumulation in cv.” What do you mean here? Incomplete sentence?

Line 190, inducing

Line 197, grammar, resulted in increases…

Line 198, our results (these are your results?)

Line 213, “only three genes…”, three out of four is not too bad, why “only”?

Lines 211-216, very long sentence needs to be rewritten for clarity.

Line 216, italics for gene names, TAA1/TAR

Line 221, “wherenFaYUC11” ?

Line 222, grammar, Despite their low expression….

Line, 234, grammar, which decrease…

Line 236, grammar, is required for the setting… What do you mean by “setting”?

Line 237, grammar, auxin transporters was reported. Expression or protein?

Line 249, grammar, “storage or inhibition of auxin signaling pathways”. The auxin signaling pathway is not stored! You mean stored IAA. Please rewrite to clarify.

Line 262, grammar, ..its transcript levels were not…

Line 304, grammar, methods showed an increase

Line 311, Vvi-miRNAs?

Line 317, Gas, GAs

Line 328, transcript levels (amounts) or protein?

Line 334-335, grammar, “is also go along with” please correct. Something like, “accompanied by”

Lines 336-341, very long sentence, consider splitting into two sentences. In addition, if there if both ethylene and respiration rate increase in raspberry fruit, are they non-climacteric?

Line 343, what do you mean by “drupelets bind to receptacle”

Lines 362-366, very long sentence, consider splitting into two sentences.

Lines 396-397, grammar

Lines 422-425, “temporal expression”? what actually happens, transcript amounts of these genes go up or down, please specify.

Line 433-435, please clarify this sentence, both treatments delayed ripening, while BL increased berry colouring and BZ decreased berry colouring?

Line 453, “maturing”, better to use “development and ripening”?

Lines 534-535, better to introduce parthenocarpy when it was previously mentioned (Line 103)

Line 539, grammar, This suggests that

Lines 561-564, do not understand this phrase, please clarify.

Lines 567-569, do not understand this phrase, please clarify.

Line 590,  grammar, participating to

Line 593-597, long and complicated sentence to understand

Author Response

Response to specific comments of Reviewer # 1:

Reviewer #1:

General Comments

The manuscript has the objective to review current knowledge about hormonal cross-talk during non-climacteric fruit developing and ripening. This would be a timely review on an interesting subject, but the manuscript needs considerable revisions. For example, the introduction needs to be reorganized to focus on the main topic, non-climacteric fruit. As the introduction is currently presented, it is not clear from the beginning which fruits are climacteric or non-climacteric. This should be addressed early in the introduction to put into context the main focus of the manuscript. Secondly, what is known about cross-talk in climacteric fruit? At least a brief summary should be included with the most recent articles on the subject referenced. Another important point, the manuscript needs extensive English editing and therefore corrections by and native English speaker is recommended during the revisions. I’ve made some corrections below noted with “grammar”, but there are many corrections to be made throughout the manuscript. In addition, need to provide full names of genes and proteins with abbreviations throughout manuscript. There are also some very long sentences that are hard to follow that need to be made more concise for clarity. Finally, the last section on hormonal cross-talk has many difficult sections to understand. This section needs to be carefully rewritten to improve clarity.

Corrections

Point 1:

Throughout manuscript (Lines 59, 291, 296, 376 and, the use of the word being is incorrect, fo example, Line 59, being the study of genes. Please check and correct throughout manuscript. Need to spell out gene and protein names with abbreviations, and then use abbreviations.

Line 97, Need to give full names for NOR, TAGLI and RIN with the abbreviations.

Lines 139-140, please give full names of genes/proteins.

Line 151, write out full name gene/protein name, VlPYL1

These suggestions were considered. As far as possible, full names and the abbreviations of gene and protein were given.

Point 2:

Line 44-Line 50, are these characteristics shared by both climacteric or nonclimacteric fruit?

All quality changes are common for both categories of fruit. It was clarified in the text.

Point 3:

Line 60-61, raspberry and blackberries are introduced here, but it is not indicated whether they are climacteric or non-climacteric.

Line 64, grape, strawberry and raspberry are mentioned here, but it is not indicated whether they are climacteric or non-climacteric.

Line 67, Glen Clova and Glen Prosen, Chilean strawberry (Fragaria chiloensis) and cultivated strawberry (Fragaria × ananassa), climacteric or non-climacteric?

Line 76, raspberry and strawberries are introduced here, but it is not indicated whether they are climacteric or non-climacteric.

As the reviewer suggested, the non-climacteric classification of raspberry, blackberry, and strawberry was clarified early in the text.

Point 4:

Line 81, the last paragraph in the Introduction and finally there is information about non-climacteric fruit. The title of the review suggests a focus on nonclimacteric fruit, so I think the introduction could be improved by focusing on the main characteristics of climacteric and non-climacteric fruit, that will lead up to the main topics of the manuscript. Examples of non-climacteric fruit discussed in the manuscript should be mentioned early in introduction.

Line 85, what about respiration, are there differences between climacteric and non-climacteric fruit?

As the reviewer suggests, a description of climacteric and non-climacteric fruit was done early in the text.

Point 5:

Line 103, “normal fruit growth even in the absence of fertilization”, you mean parthenocarpy?

Lines 534-535, better to introduce parthenocarpy when it was previously mentioned (Line 103)

As the reviewer suggested, the parthenocarpy description was better introduced in Line 103.

Point 6:

Lines 121-122, what affect, increase gene expression?

Line 123, do you mean genes for L-ascorbic acid and folic acid are also increased? Please clarify.

As the reviewer suggested, the increase in these genes was clarified.

Point 7: 

Lines 171-172, “accumulation in cv.” What do you mean here? Incomplete sentence?

These suggestions were considered, and the sentence was completed in the text.

 Point 8:

Line 311, Vvi-miRNAs?

As the reviewer suggested, the miRNAs term was clarified in the text.

 Point 9:

Line 328, transcript levels (amounts) or protein?

As the reviewer suggested, the transcript levels were clarified in the text.

Point 10:

Lines 336-341, very long sentence, consider splitting into two sentences. In addition, if there if both ethylene and respiration rate increase in raspberry fruit, are they non-climacteric?

As suggest the reviewer, this point was clarified in the manuscript.

The classification of several fruits, “melon, kiwi, strawberry, raspberry”, as either climacteric or non-climacteric is not obvious was added. The postharvest analysis in raspberry indicated a non-climacteric classification of raspberry. However, new data shows an ethylene production and respiratory rate according to climacteric classification.

 Point 11:

Lines 422-425, “temporal expression”? what actually happens, transcript amounts of these genes go up or down, please specify.

As suggest the reviewer. The sentence was improved for better understand.

  Point 12:

Line 38, “Evolutionarily, have been reported that…” is grammatically incorrect. Please correct.

Line 134, “g Asubstantial” ? Please correct.

Line 138, grammar, have been

Line 141, binding, not biding

Line 145, grammar and spelling, “), activated the expresion", please correct

Line 148, grammar, becomes high

Line 150, grammar, have recently

Line 156, also improved

Line 190, inducing

Line 197, grammar, resulted in increases…

Line 198, our results (these are your results?)

Line 213, “only three genes…”, three out of four is not too bad, why “only”?

Lines 211-216, very long sentence needs to be rewritten for clarity.

Line 216, italics for gene names, TAA1/TAR

Line 221, “wherenFaYUC11” ?

Line 222, grammar, Despite their low expression….

Line, 234, grammar, which decrease…

Line 236, grammar, is required for the setting… What do you mean by “setting”?

Line 237, grammar, auxin transporters was reported. Expression or protein?

Line 249, grammar, “storage or inhibition of auxin signaling pathways”. The auxin signaling pathway is not stored! You mean stored IAA. Please rewrite to clarify.

Line 262, grammar, ..its transcript levels were not…

Line 304, grammar, methods showed an increase

Line 317, Gas, GAs

Line 334-335, grammar, “is also go along with” please correct. Something like, “accompanied by”

Line 343, what do you mean by “drupelets bind to receptacle”

Lines 362-366, very long sentence, consider splitting into two sentences.

Lines 396-397, grammar

Line 433-435, please clarify this sentence, both treatments delayed ripening, while BL increased berry colouring and BZ decreased berry colouring?

Line 453, “maturing”, better to use “development and ripening”?

Line 539, grammar, This suggests that

Lines 561-564, do not understand this phrase, please clarify.

Lines 567-569, do not understand this phrase, please clarify.

Line 590, grammar, participating to

Line 593-597, long and complicated sentence to understand

As suggest the reviewer, all these grammar problems were edited and no clear sentences were rewritten.

The authors really thank to Reviewer #1

Reviewer 2 Report

Dear Authors,

I carefully assessed the manuscript entitled “Recent advances in hormonal regulation and cross-talk during non-climacteric fruit development and ripening”. The authors make attempt to summarize the physiology underlying the ripening and developmental processes in non-climacteric fleshy fruit, with particular regards to the hormonal cross-talk taking place during that processes-

The paper is written in a proper English and presents in a clear way the most recent discoveries on the field.

The manuscript is worthy and can be considered useful for the reader. Nevertheless there are some improvements that the authors can do in order to increase the readability of the manuscript:

1.     Line 36 change “resulting from the receptacle tissue or from enlargement of the sepals” to “ resulting from the accessory tissue exterior to the carpels”.

2.     Line 134 “.g Asubstantial” should be “. A substantial”.

3.     Line 175 “Tomato fruit s” should be “Tomato fruits”

4.     Line 245-265 You should try to include briefly in your discussion also the crosstalk exiting between GH3 genes and ILL gene family (IAA-amino acid hydrolase), as explained by Fortes et al. (2015 – ref 49 in the manuscript). A similar crosstalk on the auxin homeostasis exists also in the climacteric fruit, playing a role in the climacteric onset (Tadiello et al., 2016 – DOI 10.1186/s12870-016-0730-7).

5.     Line 604, more recently the role and function of RIN have been re-evaluated. (Ito et al., 2017 – DOI 10.1038/s41477-017-0041-5 and Jovannoni, 2017- DOI 10.1038/s41477-017-0062-0). The authors should be aware of that and outline this elements in the discussion.

6.     If possible, it would be really helpful to sketch an additional schematic figure summarizing the complex interaction between hormones taking place during the non climacteric ripening (similar to DOI 10.1016/j.molp.2014.11.006, fig 3 - DOI 10.1038/srep43364, fig 10 - DOI 10.1016/j.scienta.2019.01.034, fig 2).

Author Response

Response to specific comments of Reviewer # 2:

Reviewer #2:

General Comments

I carefully assessed the manuscript entitled “Recent advances in hormonal regulation and cross-talk during non-climacteric fruit development and ripening”. The authors make attempt to summarize the physiology underlying the ripening and developmental processes in non-climacteric fleshy fruit, with particular regards to the hormonal cross-talk taking place during that processes- The paper is written in a proper English and presents in a clear way the most recent discoveries on the field. The manuscript is worthy and can be considered useful for the reader. Nevertheless there are some improvements that the authors can do in order to increase the readability of the manuscript:

Corrections

Point 1:

1. Line 36 change “resulting from the receptacle tissue or from enlargement of the sepals” to “ resulting from the accessory tissue exterior to the carpels”.

As suggest the reviewer, these sentences were changed.

Point 2:

2. Line 134 “.g Asubstantial” should be “. A substantial”.

3. Line 175 “Tomato fruit s” should be “Tomato fruits”

As suggest the reviewer, all these grammar problems among others were edited in the manuscript.

Point 3:

4. Line 245-265 You should try to include briefly in your discussion also the crosstalk exiting between GH3 genes and ILL gene family (IAA-amino acid hydrolase), as explained by Fortes et al. (2015 – ref 49 in the manuscript). A similar crosstalk on the auxin homeostasis exists also in the climacteric fruit, playing a role in the climacteric onset (Tadiello et al., 2016 – DOI 10.1186/s12870-016-0730-7).

As suggest the reviewer, antecedents about IAA-amino acid hydrolase were included in the manuscript.

Point 4:

5. Line 604, more recently the role and function of RIN have been reevaluated. (Ito et al., 2017 – DOI 10.1038/s41477-017-0041-5 and Jovannoni, 2017- DOI 10.1038/s41477-017-0062-0). The authors should be aware of that and outline this elements in the discussion.

As suggest the reviewer, new antecedents about RIN were included in the manuscript.

Point 5:

6. If possible, it would be really helpful to sketch an additional schematic figure summarizing the complex interaction between hormones taking place during the non climacteric ripening (similar to DOI 10.1016/j.molp.2014.11.006, fig 3 - DOI 10.1038/srep43364, fig 10 - DOI 10.1016/j.scienta.2019.01.034, fig 2).

As suggest the reviewer, a schematic figure summarizing the complex interaction between hormones was included (Figure 2).

The authors really thank to Reviewer #2

Reviewer 3 Report

In this review article the authors summarize the findings about phytohormonal cross-talk during non- climacteric fruit (grape, strawberry and raspberry) development and ripening. The authors described the role of auxins, cytokinins, ethylene, ABA, jasmonates, gibberellins and brassinosteroids.  I can recomend this manuscript for publication in Horticulturae after minor revisions listed below:

1)     I recommend adding a schematic model (Fig.2) of phytohormonal cross-talk in fruit ripening and development. It make it better for reader clarity.

2)     Add a subchapter 3.3 about the phytohormones cross-talk in raspberries (because in the previous text you focus on grape, strawberry and raspberry…..)

Author Response

Response to specific comments of Reviewer # 3:

Reviewer #3:

General Comments

In this review article the authors summarize the findings about phytohormonal cross-talk during non- climacteric fruit (grape, strawberry and raspberry) development and ripening. The authors described the role of auxins, cytokinins, ethylene, ABA, jasmonates, gibberellins and brassinosteroids. I can recomend this manuscript for publication in Horticulturae after minor revisions listed below:

1) I recommend adding a schematic model (Fig.2) of phytohormonal cross-talk in fruit ripening and development. It make it better for reader clarity.

As suggest the reviewer, a schematic figure summarizing the complex interaction between hormones was included (Figure 2).

2) Add a subchapter 3.3 about the phytohormones cross-talk in raspberries (because in the previous text you focus on grape, strawberry and raspberry…..)

As suggest the reviewer, subchapter 3.3 about the phytohormones cross-talk in raspberries was included

The authors really thank to Reviewer #3.

Round 2

Reviewer 1 Report

The revised manuscript (horticulturae-449187) entitled "Recent advances in hormonal regulation and cross-talk during non-climacteric fruit development and ripening" by Fuentes et al. was reviewed for publication in Horticulturae. The authors have done a good job with the revisions, but still there are some sections throughout the manuscript that need correction, particularly the English grammar. Below are some of the errors I found while reading, but there may be more, recommend text editing by a native English speaker.

Minor corrections

Line 42, …fruit involves a progression…

Line 44, …is marked by very important phase changes…

Line 50, … in changes in fruit texture… (texture cannot decrease, it is a qualitative character)

Line 143, …. results in….

Line 217-218, not sure why you talk about tomato here after introducing the phrase with non-climacteric fruit

Line 235, … to IAA’s role and relation….

Line 245, … are highly expressed during development ….

Line 289, delete this “,indeed the importance of this process during fruit ripening”

Line 303, … conjugates and in…  

Line 328-329, auxin response transcription factors (ARFs) are a class of transcriptional regulators, while RIN is a MADs-box. The RIN gene responds to auxin, but is not an ARF.

Line 490, …one of the aspects more…

Line 563, Hormone cross talk

Line 705, All these changes were…

Line 706, …significant increase of the pulling force…. …detach the berry from the and the

Line 709-710, However, the relation of these effects with some phytohormones still need to be deciphered.

Author Response

Response to specific comments of Reviewer # 1:

Reviewer #1:

Corrections

Point 1

Line 42, …fruit involves a progression…

Line 44, …is marked by very important phase changes…

Line 50, … in changes in fruit texture… (texture cannot decrease, it is a qualitative character)

Line 143, …. results in….

Line 235, … to IAA’s role and relation….

Line 245, … are highly expressed during development ….

Line 289, delete this “,indeed the importance of this process during fruit ripening”

Line 303, … conjugates and in…

Line 490, …one of the aspects more…

Line 563, Hormone cross talk

Line 705, All these changes were…

Line 706, …significant increase of the pulling force…. …detach the berry from the and the

Line 709-710, However, the relation of these effects with some phytohormones still need to be deciphered.

As suggest the reviewer, all these grammar problems among others were edited and no clear sentences were rewritten.

Point 2

Line 217-218, not sure why you talk about tomato here after introducing the phrase with non-climacteric fruit.

A sentence clarified the IAA role during the development of climacteric and non-climacteric fruit was included to introduce the phrase about tomato fruit.

Point 3

Line 328-329, auxin response transcription factors (ARFs) are a class of transcriptional regulators, while RIN is a MADs-box. The RIN gene responds to auxin, but is not an ARF.

The text does not mention that RIN is an ARF. However, the order of the paragraphs was changed to not generate confusion